# Floods and droughts: a multivariate perspective

Manuela Irene Brunner[1,2,3]

[1]WSL Institute for Snow and Avalanche Research SLF, Davos Dorf, Switzerland
[2]Institute for Atmospheric and Climate Science, ETH Zurich, Zurich, Switzerland
[3]Climate Change, Extremes and Natural Hazards in Alpine Regions Research Center CERC, Davos Dorf, Switzerland

**Correspondence:** Manuela I. Brunner (manuela.brunner@slf.ch)

**Abstract.** Multivariate or compound hydrological extreme events such as successive floods, large-scale droughts, or consecutive drought-to-flood events challenge water management and can be particularly impactful. Still, the multivariate nature of floods and droughts is often ignored by studying individual characteristics only, which can lead to risk under- or overestimation. Studying multivariate extremes is challenging because of variable dependencies and because they are even less abundant in observational records than univariate extremes. In this review, I discuss different types of multivariate hydrological extremes and their dependencies including regional extremes affecting multiple locations such as spatially connected flood events, consecutive extremes occurring in close temporal succession such as successive droughts, extremes characterized by multiple characteristics such as floods with jointly high peak discharge and flood volume, and transitions between different types of extremes such as drought-to-flood transitions. I present different strategies to describe and model multivariate extremes, and to assess their hazard potential including descriptors of multivariate extremes, multivariate distributions and return periods, as well as stochastic and large-ensemble simulation approaches. The strategies discussed enable a multivariate perspective on hydrological extremes, which allows us to derive risk estimates for extreme events described by more than one variable.

## 1 Introduction

In July 2021, a severe and widespread flood event affected Western Germany and parts of Belgium and the Netherlands where it led to numerous fatalities and considerable damage to infrastructure (Kreienkamp et al., 2021; Ibebuchi, 2022). After such exceptional flood events, we ask: 'how frequently do such events occur?' To answer this question, one can rely on frequency analyses which establish a link between the magnitude and frequency of events. Such analyses are often performed by focusing on one variable only, i.e. by taking a univariate perspective. In the case of the Germany flood, this would e.g. be flood peaks in one individual catchment. While such a focus on one variable enables the development of suitable preparedness and adaptation measures by providing magnitude and frequency estimates of extreme events, they have a major drawback: they neglect that extremes are often not univariate but multivariate phenomena, i.e. affect more than one variable. To illustrate the multivariate nature of hydrologic extremes, let's again look at the 2021 flood. This flood event was not just extreme in terms of peak discharge at one location, it was also extreme in terms of the flood volume generated. Furthermore, it affected not just one catchment but multiple catchments in Germany, Belgium, and the Netherlands. This example highlights that the multivariate nature of hydrological extremes can take multiple forms. In the case of peak discharge and volume, we are looking at an

extreme event characterized by multiple variables and in the case of multiple affected locations at a regional extreme event. These different types of multivariate extremes have in common that they involve multiple interdependent variables, which requires a multivariate perspective. In this review, I first provide an overview of different types of multivariate hydrological extremes including regional extremes, consecutive extremes, extremes with multiple characteristics, and extremes transitions. In addition, I review tools, measures, and descriptors available to describe these different types of extremes. Second, I present modeling approaches available to model extremes in a multivariate framework, such as copula models and multivariate simulation approaches. Last, I discuss challenges related to multivariate hydrological extremes, including the regionalization of multivariate extremes to ungauged basins and the assessment of future changes in multivariate extreme events.

## 2   Types of multivariate hydrological extremes

The multivariate nature of hydrological extreme events can take multiple forms (Figure 1). A first type of multivariate hydrological extremes is regional extremes that affect multiple catchments at once. The 2021 flood in Germany is an example of such a regional extreme event (Figure 1a). Regional extremes represent a challenge for emergency management because resources need to be distributed and shared across regions. A second type of multivariate hydrological extremes is consecutive extremes, i.e. several extreme events occurring in close temporal succession (Figure 1b). An example for such a consecutive extreme event is the 'multi-year' drought 2018–2020 characterized by multiple dry summers over Central Europe (Rakovec et al., 2022), which severely impacted water supply and agriculture (Stephan et al., 2021) and had severe ecological consequences such as forest die backs (Sánchez-Pinillos et al., 2022). A third type of multivariate extreme is hydrological extremes described by multiple characteristics such as flood peak and volume as in the case of the 2021 flood event in Germany (Kreienkamp et al., 2021) (Figure 1c). Such extremes, which affect multiple characteristics, challenge water management because hydraulic structures such as retention basins have to cope not just with high maximum loads but also high volumes. A fourth type of multivariate hydrologic extremes is transitions from one type of extreme event to another type of extreme event, such as drought-to-flood transitions (Figure 1d). An example for such a drought-to-flood transition event is the multi-year dry period in California (2011–2016) which was ended by a flood in 2017 (Swain et al., 2018; He and Sheffield, 2020). Such transition events can also challenge water management because regulation measures, which might be reasonable from the perspective of one type of extreme, may be less useful from the perspective of the other type of extreme (Ward et al., 2020). In the following sections, I review the state of knowledge on these four types of multivariate hydrological extremes, i.e., regional and consecutive extremes, extremes with multiple characteristics, and extremes transitions. In addition, I summarize methodological tools used to study these different types of multivariate hydrological extreme events.

### 2.1   Regional extremes

Regional extremes affect multiple locations, catchments, or river basins at (almost) the same time and are also called spatially compounding extremes (Zscheischler et al., 2020). Here, we talk about regional extremes as soon as a local perspective is

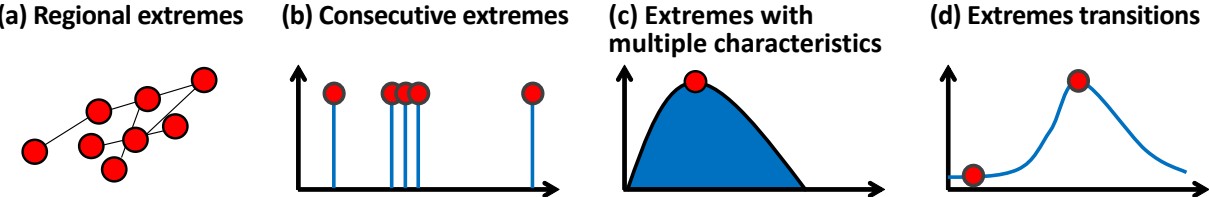

**Figure 1.** Illustration of different types of multivariate hydrological extreme events: (a) Regional extremes, (b) consecutive extremes, (c) extremes with multiple characteristics, and (d) extremes transitions.

no longer sufficient, i.e. when floods have a larger spatial extent and more than one catchment is affected, which requires a multivariate perspective.

### 2.1.1 Regional floods

Floods can occur simultaneously at multiple locations, i.e. flood occurrences can be spatially dependent (Figure 2). Such

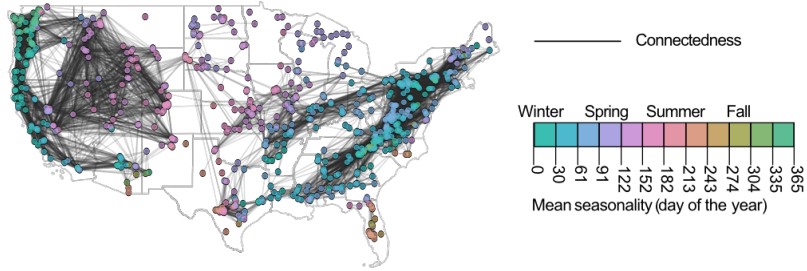

**Figure 2.** Spatial flood connectedness in the United States computed over all seasons. Links indicate stations, which have at least 10 flood events in common. Stations are colored according to the mean day of flood occurrence.

spatial dependence can be quantified using different types of measures, including pairwise measures such as the number of co-occurrences at a pair of catchments (Brunner et al., 2020a) or the correlation between flood magnitudes at a pair of catchments (Brunner and Gilleland, 2021); catchment specific measures such as the distance over which multiple catchments flood near synchronously (i.e. the flood synchrony scale; Berghuijs et al., 2019) and the expected proportion of sites in a
65 catchment's vicinity that exceed their $x$th quantile during an event in which this catchment exceeds its $x$th quantile (conditional spatial dependence; Keef et al., 2009); or event-based metrics such as flood extent (Kussul et al., 2008) and the percentage of catchments affected by flooding within a certain region (Brunner et al., 2020b) (Table 1).

Spatial dependence is related to flood magnitude to a certain degree. However, spatial dependence has been shown to increase or decrease with event magnitude when using different dependence measures. Keef et al. (2009) have shown that conditional
spatial dependence is particularly severe for moderate floods and becomes weaker as events get more extreme. That is, they showed that more extreme events are more localized than moderate floods. In contrast, Kemter et al. (2020) have shown a

positive relationship between flood magnitude and extent when using the flood synchrony scale, i.e. increasing spatial scales with increasing flood magnitude. The strength of spatial dependence also depends on location and is highly variable across catchments. Berghuijs et al. (2019) have shown that the distance over which multiple catchments flood near synchronously exceeds the size of individual catchments in Europe and shows strong regional variations, with larger floods occurring in lowland than in mountain catchments.

Regional floods are shaped by both meteorological and land surface processes, i.e. precipitation spatial dependence alone is not sufficient to explain spatial flood dependence (Brunner et al., 2020a). Regional floods often develop when a storm meets favorable antecedent conditions, such as widespread wet soils, or when multiple catchments experience synchronous snowmelt (Brunner and Dougherty, 2022). Therefore, floods are more likely to be spatially connected in mountain regions with seasonal snowmelt contributions than in lowland catchments where floods are mainly driven by precipitation (Brunner and Fischer, 2022). Besides climate, spatial flood dependence is shaped by reservoir regulation, which leads to less spatially connected floods in winter compared to unregulated conditions (Brunner, 2021).

Regional flood characteristics change over time, but the direction of change is yet unclear. Berghuijs et al. (2019) have shown historical increases in the distance over which catchments flood near synchronously for catchments in Europe. In contrast, Rupp et al. (2021) found decreases in the synchrony of flooding between snowmelt-dominated basins because of decreases in snowmelt using simulations of future streamflow. This finding is in line with results by Brunner and Fischer (2022) and Brunner and Dougherty (2022) who found stronger spatial connectedness for snowmelt-influenced regions than for rainfall-driven regions. While these studies provide first evidence for future changes of regional floods in a warming climate, the direction and magnitude of these changes needs to be quantified using further targeted modelling experiments (see Section 3.4). The spatial dependencies between flood occurrences at multiple locations need to be considered in flood hazard assessments in order to avoid risk over- or underestimation (Metin et al., 2020). Such consideration can e.g. be achieved by computing probabilities of regional flooding (Brunner et al., 2020b).

### 2.1.2 Regional droughts

Droughts are often regional phenomena, i.e. drought occurrences at different locations are dependent. Similarly to floods, such spatial drought dependence can be quantified using different types of descriptors. Using a pairwise-perspective, drought dependence can be quantified by counting the number of drought co-occurrences or the number of months under concurrent drought (Brunner and Gilleland, 2021). Taking a regional perspective, regional droughts can be described by the number of catchments affected by drought (Teutschbein et al., 2022) or by the drought extent (Hanel et al., 2018). The main part of the literature studying regional droughts and their extents focuses on meteorological rather than on streamflow droughts (Ganguli and Ganguly, 2016; Sharma and Mujumdar, 2017; Perez Arango et al., 2021; Ionita and Nagavciuc, 2021). Those studies that have assessed the spatio-temporal variation in hydrological drought extents found substantial temporal variations in the number of catchments jointly affected by drought (Hanel et al., 2018; Brunner et al., 2021b; Teutschbein et al., 2022).

Spatial drought extent is driven by different hydro-meteorological conditions including soil moisture deficits, precipitation deficits, and positive temperature anomalies. The relative importance of these different drivers varies by event and season. In

winter and spring, large scale droughts often co-occur with soil moisture and precipitation deficits, while they co-occur with positive temperature anomalies in summer (Brunner et al., 2021b). While there exist first indications that the relationships between climatic drivers and drought extent are complex, future studies should focus on the identification of atmospheric drivers of widespread streamflow droughts similar to studies that assess the link between atmospheric patterns and/or climate indices and the spatial extent of meteorological droughts (e.g. McCabe and Wolock, 2022).

Streamflow drought spatial extents have increased in the United States over time, mainly because of increases in the extent of small droughts and in temperature (Brunner et al., 2021b). Further investigations are needed to assess whether such changes can also be observed in other climate zones such as tropical, arctic, or alpine regions. The spatial extents of streamflow droughts have not just changed in the past, they are also projected to further increase in future, as demonstrated for Great Britain using climate and hydrological model simulations (Rudd et al., 2019). How such changes translate to other regions remains to be assessed using modelling experiments, which focus on reliably reproducing spatial streamflow drought extents.

### 2.1.3 Descriptors of regional extremes

A diverse range of tools can be used to quantify the spatial dependence and spatial extents of floods and droughts. These tools include areal coverage, spatial extent, conditional spatial dependence, synchrony scale, length scale, probability of regional extremes, connectedness, severity-area-frequency curves, and severity-area-duration curves (Table 1). A first category of descriptors describes the spatial extent of extreme events at an event scale. This category comprises areal coverage, i.e. the percentage of a region or river basin under extreme conditions; spatial extent, i.e. the area under extreme conditions usually derived from gridded data; and conditional spatial dependence, i.e. the expected proportion of sites in the vicinity of a specific catchment that exceed their pth quantile during an event in which this catchment exceeds its pth quantile. While these descriptors focus on describing individual events, a second group of descriptors summarizes the behavior of regional extremes at a catchment scale. For example, the synchrony scale measures over which distance around a catchment, multiple rivers experience flooding at the same time. A third group of metrics comprises metrics that summarizes regional relationships in extremes occurrence e.g. through a semivariogram or more specifically the length scale (i.e. the range of the semi-variogram) or the probability of regional extremes, i.e. the probability that a certain percentage of catchments within a region is jointly under extreme conditions. A fourth group of metrics includes pairwise measures such as connectedness determined either based on the number of co-occurrences at a pair of catchments or on the correlation between flood magnitudes at a pair of catchments. A last group of descriptors are frequency or duration curves, e.g. severity-area-frequency curves or severity-area-duration curves. Depending on which metric is chosen to describe regional extremes, the results of an analysis will differ. For example, change assessments may find different changes in regional extremes when looking at pairwise relationships than when focusing at the event-scale.

### 2.2 Consecutive extremes

Consecutive extremes occur in close temporal succession in the same catchment or region and are also referred to as temporally compounding extremes (Zscheischler et al., 2020). Such temporal clustering behavior is illustrated in Figure 3, which shows

**Table 1.** Metrics used to describe regional floods and droughts

| Metric | Description | References | Application |
|---|---|---|---|
| Areal coverage | Percentage of area/catchments under extreme conditions | Rossi et al. (1992), Hannaford et al. (2010), Hanel et al. (2018), Brunner et al. (2021b) | Droughts |
| Spatial extent | Area under extreme conditions derived from gridded data | Kussul et al. (2008), Rudd et al. (2019) | Floods and droughts |
| Conditional spatial dependence | Expected proportion of sites in the vicinity $D$ of a specific catchment that exceed their $p$th quantile during an event in which this catchment exceeds its $p$th quantile | Keef et al. (2009) | Floods |
| Synchrony scale | Distance over which multiple rivers flood near synchronously | Berghuijs et al. (2019) | Floods |
| Length scale | Range of semi-variogram | Touma et al. (2018) | Extreme precipitation |
| Connectedness | Network degree, i.e. number of catchments a catchment has co-experienced extreme events with | Brunner et al. (2020a), Brunner and Gilleland (2021) | Floods and low flows |
| Probability of regional extremes | Probability that a certain percentage of catchments within a region is jointly under extreme conditions | Brunner et al. (2020b) | Floods |
| Severity-area-frequency curves | Relationship of specific severity (deficit) and area coverage for different return periods | Henriques and Santos (1999), Hisdal and Tallaksen (2003) | Droughts |
| Severity-area-duration curves | Relationship between drought severity (deficit) and area coverage for different drought durations | Andreadis et al. (2005), Sheffield et al. (2009) | Droughts |

time series of drought occurrences for two example catchments in different hydro-climates. The first catchment shows temporal
drought clustering at seasonal time scales (Figure 3a), meaning that droughts are likely to occur in subsequent seasons. The
second catchment shows temporal clustering at longer, i.e. multi-annual time scales (Figure 3b), meaning that the catchment is
affected by droughts in regular multi-annual intervals.

### 2.2.1 Consecutive floods

Flood events cluster in time, i.e. flood-rich periods in which floods are more common alternate with flood-poor periods in which
floods are rare (Villarini et al., 2013; Mediero et al., 2015; Merz et al., 2016; Gu et al., 2016; Liu and Zhang, 2017; Wang et al.,
2020). In Europe or China for example, many catchments show temporal clustering for moderate floods at time scales of one
to a few years (Merz et al., 2016; Gu et al., 2016; Lun et al., 2020). However, the strength of temporal clustering decreases
substantially with time scale and with an increasing flood threshold (Lun et al., 2020). The temporal flood clustering behavior
to some degree also depends on the region. For example, catchments in the Atlantic and Continental regions of Europe are
150 more prone to temporal flood clustering than catchments in Scandinavia (Mediero et al., 2015).

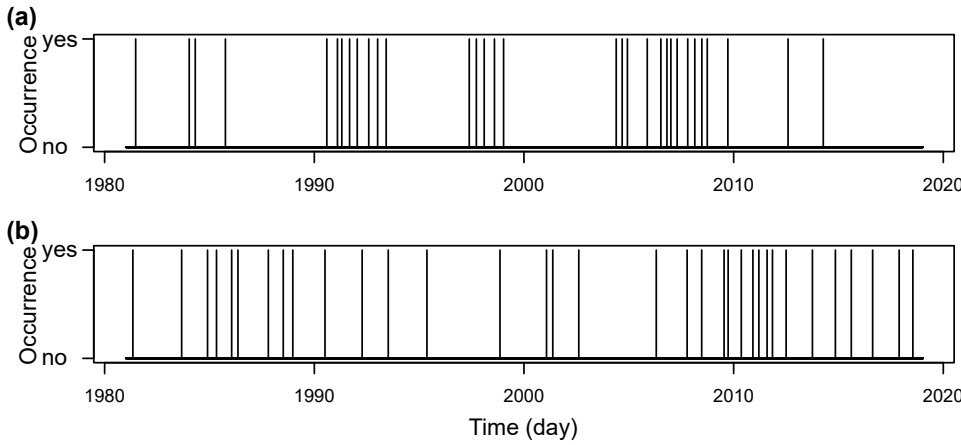

**Figure 3.** Temporal hydrological drought variability (droughts were here defined using a variable threshold at the 15th flow percentile) : (a) temporal drought occurrence in the Riss catchment at Warthausen (Austria) and (b) temporal drought occurrence in the Little Pee Dee catchment at Galivants Ferry (United States).

Flood-rich periods with temporally clustered events are related to climate. Blöschl et al. (2020) and Brönnimann et al. (2022) have e.g. shown for Europe that historic flood-rich periods occurred under colder than normal climate conditions. Similarly, Villarini et al. (2013), Gu et al. (2016), Liu and Zhang (2017) have shown for catchments in Iowa, China, and Australia, respectively, that the flood clustering behavior is influenced by large-scale climate indices. The pronounced link between climate and the temporal flood clustering behavior suggests that future changes in temperature and oscillation patterns may lead to changes in temporal flood clustering. How the temporal flood clustering behavior changes across different climate zones in a warming climate still needs to be investigated using simulation-based studies. Such simulation-based studies require the development of modeling approaches that reliably represent the temporal clustering behavior of floods.

### 2.2.2 Consecutive droughts

Drought events can occur successively or cluster in time as highlighted by studies looking at the occurrence of multi-year droughts and studies assessing the temporal clustering behavior of droughts. A first body of literature provides evidence for the occurrence of multi-year droughts both from a meteorological and hydrological perspective. The occurrence of multi-year precipitation deficits has for example been documented for France (Vidal et al., 2010), Central Europe (Moravec et al., 2021), and the United States (Goodrich, 2007; Diffenbaugh et al., 2015; Abatan et al., 2017; Bales et al., 2018) and the occurrence of multi-year streamflow deficits for different parts of Europe (Parry et al., 2012; Folland et al., 2015; Hanel et al., 2018; Brunner and Tallaksen, 2019) and Chile (Alvarez-Garreton et al., 2021). A second body of literature shows that both meteorological and hydrological drought occurrences are highly variable in time with alternations between drought-rich and drought-poor periods at multi-year (Moreira et al., 2015; Noone et al., 2017; Yue et al., 2021), decadal (Ionita et al., 2012; Tong et al., 2018;

Barker et al., 2019), and multi-decadal time scales (Tanguy et al., 2021). However, some other studies also provide contrasting evidence by showing a lack of cyclicity in precipitation deficits (Pelletier and Turcotte, 1997; Bunde et al., 2013).

Brunner and Tallaksen (2019) have shown that catchments experiencing multi-year droughts are mostly characterized by a rainfall-dominated flow regime, while catchments with melt-dominated flow regimes are generally not affected by multi-year droughts. In addition, Brunner and Stahl (2023) have shown that the temporal clustering of hydrological droughts is substantially more pronounced than the clustering of precipitation deficits. That is, climatic drivers are insufficient to explain the temporal clustering of hydrological droughts, suggesting that additional land-surface processes such as snow storage or the absence thereof, seasonal and inter-annual groundwater level variations, temporal soil moisture variability, or fluctuations in glacier-melt contributions are needed to explain hydrological drought clustering behavior. Catchments prone to temporal hydrological drought clustering are often arid and lack substantial snow storage (Brunner and Stahl, 2023). As a consequence, changes in the number of catchments showing temporal hydrological drought clustering may be expected in a warming climate because of increases in aridity and decreases in snowmelt. Similarly, multi-year droughts may become more frequent in a future climate as flow regimes transition from snow-dominated to rainfall-dominated (Brunner and Tallaksen, 2019). Detailed modeling assessments are needed to show how the probability of occurrence of multi-year droughts and the temporal-clustering behavior of droughts are going to change in the future. Such assessments require an adequate representation of temporal streamflow dependencies.

### 2.2.3 Descriptors of consecutive extremes

The persistence and periodic features of hydrological extreme events have been documented using a range of measures including the Hurst exponent, power spectra derived using the Fourier transform, dry-to-dry transition probabilities, and others (Table 2). A very simple measure to characterize consecutive extremes is the number of consecutive events, e.g. the number of successive extreme months/years. Also related to individual events, one can compute extreme event transition probabilities, i.e. the probability of observing a subsequent extreme event given that an extreme event has occurred in the previous time unit (e.g. week/month/year). Instead of focusing on events, the temporal persistence of extremes can be summarized for entire time series of extreme events, for example by the Hurst exponent, which measures the long-term memory of a time series, or the average power spectrum, i.e. the average power over all frequencies after the Fourier transform. In addition, consecutive extreme events can be described by measures that characterize the temporal clustering behavior of extreme events including the dispersion index, which quantifies the departure of an observed process from a homogeneous Poisson process, Ripley's $K$, which counts the average number of extreme events in the temporal neighborhood of extreme events, and Kernel estimation, which estimates the time variation of extreme event counts as a smooth function of time. Another possibility to describe consecutive extremes is to identify flood/drought-rich and -poor periods using scan statistics. That is, unusual periods in the observations that are inconsistent with the assumption of independent and identically distributed random variables, i.e. periods encompassing very few or very many events, are identified with a moving window approach. If it is not just of interest to describe consecutive extremes but to identify their drivers, one can rely on cox regression models, which examine the dependence of the rate of occurrence of extremes on covariate processes, e.g. different types of teleconnection patterns. The choice of a specific descriptor

**Table 2.** Metrics used to describe consecutive floods and droughts

| Metric | Description | References | Application |
|---|---|---|---|
| Number of consecutive events | Count of the number of successive extreme events/years | Hanel et al. (2018), Brunner and Tallaksen (2019) | Droughts |
| Extreme event transition probabilities | Probability of observing a subsequent extreme event given that an extreme event has occurred in the previous time unit (e.g. month) | Moon et al. (2018) | Droughts |
| Hurst exponent | Measure of the long-term memory of a time series | Hurst (1956), Tatli (2015), Noorisameleh et al. (2021) | Droughts |
| Average power spectrum | Average power over all frequencies after the Fourier transform | Pelletier and Turcotte (1997) | Droughts |
| Dispersion index | Quantifies the departure from a homogeneous Poisson process | Vitolo et al. (2009), Mediero et al. (2015), Merz et al. (2016) | Floods and droughts |
| Ripley's $K$ | Measures the average number of extreme events in the temporal neighborhood of extreme events | Ripley (1981), Dixon (2013), Tuel and Martius (2021), Tuel et al. (2022) | Extreme precipitation, floods, and droughts |
| Kernel estimation | Estimates the time variation of extreme event counts as smooth functions of time | Cowling et al. (1996), Mudelsee et al. (2003), (Merz et al., 2016) | Floods |
| Scan statistics | Maximum number of observed counts in a series of overlapping sliding windows | Lun et al. (2020) | Floods |
| Cox regression model | Cox processes are Poisson processes with a randomly varying rate of occurrence. Cox regression models can be used to examine the dependence of the rate of occurrence on covariate processes | Villarini et al. (2013) | Floods |

will depend on the specific research question or application, i.e. on whether one would like to test for clustering significance, in which case Ripley's K or the dispersion index can be used, or whether one would like to identify specific periods particularly abundant in extremes occurrence, in which case scan statistics or Kernel estimation can be used, or one would like to explain temporal dependence, in which case one can rely on cox regression models.

## 2.3 Extremes with multiple characteristics

Droughts and floods are characterized by multiple characteristics such as deficit and duration or peak discharge and flood volume, respectively (see Figure 4). These characteristics can be mutually interdependent as illustrated by some examples in Figure 5 for different drought and flood characteristics.

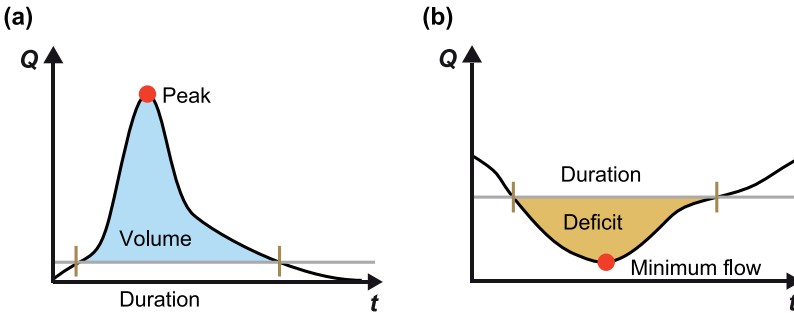

**Figure 4.** Illustration of flood and drought characteristics: (a) floods: peak discharge, volume, and duration; (b) droughts: minimum flow, deficit, and duration.

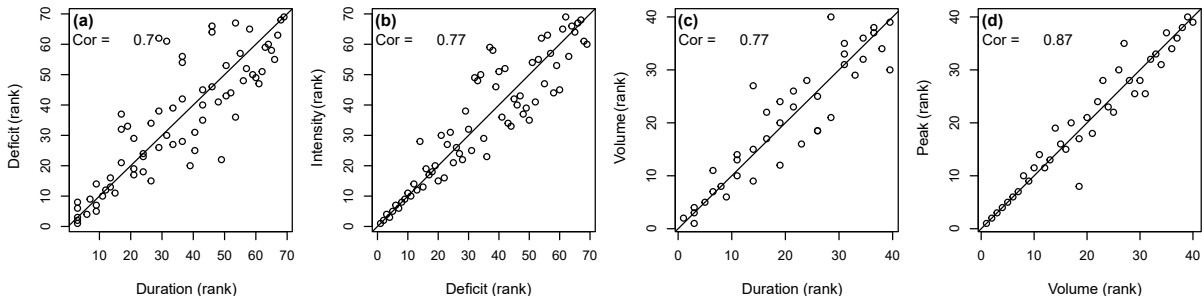

**Figure 5.** Illustration of the relationship between different drought and flood variables for the Fish river in Maine, United States: (a) drought duration and deficit, (b) drought deficit and intensity, (c) flood duration and volume, and (d) flood volume and peak discharge.

### 2.3.1 Floods

Floods are characterized by multiple characteristics including peak discharge, volume, and duration (Figure 4a), which are interdependent (Mediero et al., 2010; Serinaldi and Grimaldi, 2011). For example, flood duration and volume or flood volume and flood peak show strong correlations (Figure 5), i.e. they show bivariate dependence. These variable relationships vary
with the flood generation process, e.g. flash-floods, short-rain floods, long-rain floods, and rain-on-snow floods show different forms and strengths of variable dependence (Renard and Lang, 2007; Szolgay et al., 2015; Brunner et al., 2017). Because of such variations in variable dependence with flood generation processes, variable dependence also varies between low- and high-elevation catchments (Gaál et al., 2015). For Austrian catchments, Gaál et al. (2015) found weaker variable dependence in Alpine than in lowland catchments because of a mix of flood generation processes. In addition to elevation, variable dependence
has also been shown to vary with catchment size. Using a global dataset, Rahimi et al. (2021) have shown that the strength of variable dependence increases with the catchment area. However, overall, variable dependence seems to be more strongly related to climatic factors than to physiographic factors (Gaál et al., 2015). Because of the link between climatic flood drivers and variable dependence, the strength of variable dependence is changing in a warming climate. For example, Bender et al.

(2014) found an increase in the dependence between flood volume and peak discharge for the Rhine river and Ben Aissia et al. (2014) detected decreases and increases in such dependence for two catchments in Québec. These temporal change patterns in variable dependence are spatially heterogeneous and cannot be explained by one hydro-meteorological driver alone. Instead, changes in variable dependence are the result of an interplay between changes in precipitation, snowmelt, and soil moisture, resulting in dependence increases in some and dependence decreases in other regions (Brunner et al., 2019c). The interdependencies between different flood variables, and their potential future changes need to be considered in multivariate hazard and climate impact assessments. That is, flood frequency analyses need to consider variable dependencies if multiple variables are of interest for the application. For example, the dependence between peak and volume should be considered when deriving flood estimates for hydraulic design.

### 2.3.2 Droughts

Similarly to floods, droughts can be described by different characteristics including drought intensity, deficit, and duration (Figure 4b), which are also interdependent (Shiau, 2006; Lee et al., 2013; Salvadori and Michele, 2015; Brunner et al., 2019d). Such bivariate interdependence is e.g. found for drought deficit and duration or drought deficit and intensity (Figure 5a,b). The strength of dependence varies with climate (Van Loon et al., 2014). Drought deficit increases most strongly with duration in cold seasonal climates because snow accumulation during winter prevents the recovery from summer drought and in monsoonal, Savannah, and Mediterranean climate zones where summer droughts continue into the winter (Van Loon et al., 2014). This relationship between drought variable dependence and climate suggests that the variable interdependence may change in a warming climate. How climate change specifically affects the dependence between different pairs of variables needs to be assessed using targeted modelling experiments focusing on an accurate representation of variable dependencies in hydrological models.

### 2.3.3 Descriptors of extremes with multiple characteristics

The interdependencies between multiple characteristics of hydrological extreme events can be assessed using various dependence measures, including different correlation and tail dependence measures focusing on bivariate variable relationships (Table 3). Linear relationships can be quantified using Pearson's correlation coefficient while non-linear relationships can be described using Spearman's or Kendall's rank correlation coefficients. If the focus is not on the bulk of the distribution but on its tails, one can use the extremal dependence coefficient, which describes the probability of one variable being extreme given that the other one is extreme.

## 2.4 Extremes transitions

Consecutive drought and flood periods can seriously challenge water and emergency management because of trade-offs between long-term water storage and short-term flood control (Di Baldassarre et al., 2017; He and Sheffield, 2020) and substantial effects on water quality (Mosley, 2015; Pulley et al., 2016). Recent examples of such events include the transition from a very

**Table 3.** Metrics used to describe hydrological extremes with multiple characteristics

| Dependence measure | Description | References | Application |
|---|---|---|---|
| Pearson's correlation coefficient | Measure of linear correlation between two data samples | Edwards (1976) | Droughts and floods |
| Spearman's rank correlation coefficient | Measure of rank correlation between two data samples | Spearman (1904); Genest and Favre (2007) | Droughts and floods |
| Kendall's rank correlation coefficient | Measure of rank correlation between two data samples | Kendall (1937); Genest and Favre (2007) | Droughts and floods |
| Extremal dependence/tail dependence coefficient | Probability of one variable being extreme given that the other one is extreme | Coles et al. (1999); Coles (2001) | Droughts and floods |

dry spring in 2017 to extremely wet conditions in July in several parts of Germany (Becker et al., 2017), the multi-year dry period in California (2011–2016) which was ended by a flood in 2017 (Swain et al., 2018; He and Sheffield, 2020), or the dry 2010–2012 period in the UK that ended with record summer rainfall (Marsh et al., 2013).

### 2.4.1 Droughts to floods

Studies looking at transitions from dry to wet periods mainly focus on transitions in meteorological states, i.e. on transitions
from negative to positive precipitation or moisture anomalies (Yang et al., 2013; Liu et al., 2018; Shi et al., 2021; Ansari and Grossi, 2022). These meteorological studies indicate large spatial variability in dry-to-wet period transition times ranging from a few months to multiple years (De Luca et al., 2020). In contrast, little is known about consecutive hydrological drought–flood events, i.e. transitions between extremes in streamflow data. For the Amazonas River, Espinoza et al. (2012) studied the abrupt transition from an extreme drought in September 2010 to very high discharge in April 2011 and Parry et al. (2016) studied
drought termination for river basins in the UK. Still, little is known about the atmospheric and land-surface conditions that lead to rapid drought to flood transitions and about how transition times and characteristics vary in space and time. Further research is needed in order to better understand the variations of transition times across hydro-climates and the hydro-climatic drivers of rapid drought–flood transitions. Studies looking at future changes in transitions between dry and wet meteorological states suggest more frequent and rapid transitions between wet and dry extremes (Chen and Ford, 2022). Hydrological simulation
experiments are needed to assess how these changes in transitions from dry to wet states translate into changes in transitions from hydrological droughts-to-floods. The possibility of rapid drought–flood transitions under both current and future climate conditions needs to be integrated in disaster risk reduction strategies (Ward et al., 2020).

**Table 4.** Metrics used to describe transitions between extreme events

| Transition measure | Description | References | Application |
|---|---|---|---|
| Transition time | Time between dry and wet periods | De Luca et al. (2020); Chen and Ford (2022) | Dry to wet conditions |
| Transition frequency | Frequency of transitions between dry and wet periods | Chen and Ford (2022) | Dry to wet conditions |

### 2.4.2 Descriptors of extremes transitions

The transitions between dry and wet periods have been described using transition times and transition frequencies as summarized in Table 4. The transition time describes the time elapsing between dry and wet periods while the transition frequency describes the frequency of transitions between dry and wet periods.

## 3 Modeling multivariate extremes

Assessments of the frequency and magnitude of multivariate hydrologic extreme events are facilitated by various tools and approaches such as describing multivariate phenomena with suitable univariate metrics, bivariate distributions and return period definitions, multivariate distributions, multivariate stochastic simulation approaches, and hydrological models.

### 3.1 Univariate metrics for multivariate extremes

Different approaches have been developed to quantify the frequency of multivariate extremes. The easiest work around for dealing with multivariate extremes is to describe the complex phenomena with a suitable univariate descriptor, such as describing regional floods by flood extent. Such univariate descriptors can be used in a univariate frequency analysis to determine the frequency and magnitude of events. Such a univariate frequency analysis first defines a sample of extreme events using either a block maxima/minima or a peak-over-threshold/threshold-level approach (Meylan et al., 2012). Second, it fits a suitable theoretical distribution to the sample of extreme events. In the case of block maxima, one usually works with a Generalized Extreme Value (GEV) distribution and in the case of threshold exceedances with a Generalized Pareto distribution (GPD) (Coles, 2001). The goodness-of-fit of the distribution chosen is assessed using a test for extreme values such as the Anderson–Darling or Cramér-von-Mises test (Laio, 2004). Once a suitable distribution has been identified, one can use the probability distribution function to determine the probability of occurrence of a certain event or the quantile function to determine the magnitude of an event with a certain non-exceedance probability or return period (Figure 6). The relationship between the non-exceedance probability $p$ and the corresponding return period $T$ is expressed as follows:

$$T = \mu/(1-p), \tag{1}$$

where $\mu$ is the mean inter-arrival time between two successive events, which is defined as one divided by the number of flood occurrences per year (Gumbel, 1941; Salvadori and De Michele, 2010; Brunner et al., 2016). Using this relationship, one can answer questions such as 'how often does an extreme event with a certain magnitude occur' or 'how big is an event with a certain return period'.

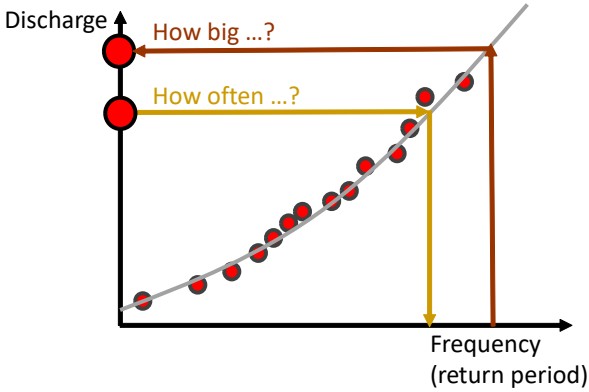

**Figure 6.** Illustration of the relationship between extreme event frequency and magnitude.

## 3.2    Bivariate distributions and return periods

In many cases, however, univariate descriptors of multivariate extremes as described above do not exist, e.g. when we are interested in floods characterized by multiple variables such as magnitude, volume, and duration. Because multivariate definitions of return periods are difficult to establish, one often tries to break down the problem to bivariate relationships, for which bivariate distributions and return period definitions exist. The joint distribution of variables that are interdependent can be represented using bivariate distributions such as the bivariate generalized extreme value distribution (Coles, 2001) or copula models, which

allow for a more flexible representation of different variable-dependence structures and different univariate distributions for the margins (Genest and Favre, 2007). The copula approach roots in the representation theorem by Sklar (1959), which states that the joint cumulative distribution function $F_{XY}$ of a pair of continuous random variables $(X, Y)$ at $(x, y)$ can be expressed by

$$F_{XY}(x, y) = C(F_X(x), F_Y(y)), x, y \in \mathbb{R}, \tag{2}$$

where $F_X(x)$ and $F_Y(y)$ are realizations of the marginal distributions of $X$ and $Y$ whose dependence is modeled by a copula $C$ (Nelsen, 2006; Joe, 2015). This copula approach allows one to select an appropriate model for the dependence between $X$ and $Y$ independently from the choice of the marginal distributions. In order to identify a suitable copula for a pair of variables, five steps have to be taken:

     1. quantify the strength of dependence and evaluate the form of dependence between the variables using rank-based corre-

lation measures and dependence plots (Genest and Favre, 2007).

2. choose a number of copula families.

3. estimate the copula parameters for each copula family.

4. perform goodness-of-fit tests to exclude unsuitable copulas (Genest et al., 2009).

5. choose one of the admissible copulas using selection criteria such as the Akaike or Bayesian information criterion.

For an introduction to copulas with application examples, the reader is referred to Genest and Favre (2007) and for detailed theoretical introductions to Nelsen (2006) and Joe (2015).

Such bivariate distributions are needed to compute return periods in a bivariate context, e.g. when hydraulic design relies on two variables such as peak discharge and flood volume. In the univariate setting, the return period $T$ is uniquely defined as described by Equation 1. In the bivariate and more generally the multivariate setting, the definition of the return period of an observed event is not unique. Instead, one has to choose one out of several definitions depending on the problem at hand (Serinaldi, 2015). In a multivariate framework, the return period can be defined as the return period $T_D$ of a "dangerous" event as

$$T_D = \frac{\mu}{\Pr[X \in D]}, \tag{3}$$

where $D$ is a set of events defined to be dangerous according to some reasonable criterion and $\Pr[X \in D]$ is the probability that the random variable $X$ lies in this dangerous region $D$. In a multivariate setting, $D$ can be defined in different ways depending on the application at hand, e.g. using the conditional probability distribution, joint probability distributions, or the Kendall's distribution (Gräler et al., 2013; Brunner et al., 2016). These distributions are typically expressed using bivariate copula models. For example, if the definition of dangerous events spans all those events where the two variables (e.g. peak discharge and flood volume) jointly exceed a certain threshold, one would use the joint 'AND' return period definition. This joint 'AND' return period $T(u,v)$ is using a copula $C$ expressed as

$$T(u,v) = \frac{\mu}{1 - u - v + C(u,v)}, \tag{4}$$

where $u$ and $v$ are realizations of $U$ and $V$, i.e. uniform representations of $F_X$ and $F_Y$. An alternative to this joint return period definition is the Kendall return period, i.e. the mean inter-arrival time of dangerous events (events more critical than the design event) (Salvadori et al., 2011). The separation between dangerous and non-dangerous events is made based on the Kendall distribution function $K_C$

$$K_C(t) = P[C(u,v) \leq t], \tag{5}$$

where $t$ is the critical probability level. The probability level $t$ corresponding to the design return period $T_K$ can be calculated from the inverse of the 2-D Kendall distribution function as

$$T_K = \frac{\mu_T}{1 - K_C(t)}. \tag{6}$$

For an overview of more alternative bivariate return period definitions, the reader is referred to Gräler (2014) or Brunner et al. (2016). Such bivariate return period definitions can be used to quantify the return period of events characterized by two variables, e.g. droughts described by drought deficit and duration or floods described by flood peak and volume (Salvadori, 2004; Serinaldi and Grimaldi, 2011; Serinaldi, 2016; Brunner et al., 2017, 2019d). However, return periods are difficult to generalize to higher than two-dimensional data (Gräler et al., 2013). An exception are 3-dimensional data for which the Kendall

return period can also be computed by determining the corresponding probability level $t$ (Salvadori et al., 2011).

## 3.3   Multivariate distributions

Different models for multivariate extremes have been proposed in the literature, including multivariate distributions such as the logistic model (Kotz and Nadarajah, 2000), conditional exceedance models (Heffernan and Tawn, 2004; Neal et al., 2013; Keef et al., 2013), the multivariate skew-$t$ distribution (Ghizzoni et al., 2010, 2012), hierarchical Bayesian models (Yan and

Moradkhani, 2015), max-stable models (Ribatet, 2013), the multivariate generalized Pareto distribution (Rootzén and Tajvidi, 2006; Rootzén et al., 2018), and copula models such as pair-copula constructions (Gräler, 2014; Schulte and Schumann, 2015; Bevacqua et al., 2017), factor copula models (Lee and Joe, 2017), vine copulas (Bedford and Cooke, 2002; Gräler et al., 2013), chi-square copulas (Bárdossy, 2006; Quessy et al., 2016) or the Fisher copula (Favre et al., 2018; Brunner et al., 2019b). Classical multivariate distributions such as the logistic model, have mostly been defined for the bivariate or trivariate

case because the complexity linked to the solution of multivariate problems increases strongly with the dimension (Kotz and Nadarajah, 2000). This dimensionality problem can be overcome by using conditional exceedance models as proposed by Heffernan and Tawn (2004), which can be applied to phenomena of any dimension, e.g. to model spatial extremes (Keef et al., 2013; Neal et al., 2013). In such a spatial extremes context, these models are defined in terms of the statistical distribution of a variable (e.g. streamflow) at a set of locations conditional on the variable exceeding a certain threshold at one of these

locations. Applications are not limited to spatial extremes and could also be extended to extremes with multiple characteristics by quantifying the conditional distribution of one variable (e.g. flood peak) being extreme given that another variable (e.g. flood volume) is high (Salvadori et al., 2014). However, in order to account for the full range of possible models, the use of conditional exceedance models requires the fitting of several models (e.g. by conditioning on each variable once). Multivariate distributions of higher dimension also exist both for componentwise maxima and threshold exceedances. Max-stable distributions arise

from the limiting behavior of vectors of componentwise maxima (block maxima) (Segers, 2012; Ribatet, 2013) and there exist a number of parametric max-stable models, e.g. Brown–Resnick processes, the Smith model, or the Hüsler-Reiss model (Davison et al., 2012). Max-stable process models have e.g. been used to model the spatial dependence of rainfall extremes (Davison et al., 2012; Le et al., 2018). Similarly, multivariate generalized Pareto distributions result from the limit distributions of exceedances over multivariate thresholds of different variables (Rootzén and Tajvidi, 2006; Rootzén et al., 2018; Kiriliouk

et al., 2019). These multivariate generalized Pareto distributions can be applied to a wider range of applications than max-stable models because they do not require the definition of pairwise extremes. Another flexible alternative to max-stable models are multivariate copula models such as vine copulas which extend to higher than two to three dimensions (Bedford and Cooke,

2002; Gräler et al., 2013). Vine copulas construct high-dimensional copulas by mixing conditional bivariate copulas in a stagewise procedure, i.e. by modeling pairwise dependencies with bivariate copulas (Gräler et al., 2013).

## 380 **3.4 Simulation of multivariate extremes**

Multivariate extreme events are even less abundant in observational records than univariate extremes. This lack of data challenges frequency analysis because reliable distribution fitting requires sufficiently large datasets. To overcome the problem of a limited sample size, different simulation approaches have been proposed, which enable simulating long time series or large event sets. These simulation approaches include statistical and physically-based models. Both types of approaches aim 385 to generate large samples of data with similar distributional and spatio-temporal characteristics as the limited observed data. Such large simulation ensembles can be used to refine water management plans, or to develop suitable adaptation strategies to drought and flood events.

There exists a variety of stochastic modeling approaches which differ in their capability of representing distributional and/or temporal characteristics of hydrological data. The most commonly used direct stochastic simulation approaches, i.e. approaches 390 that directly simulate streamflow using a stochastic model, belong to the two classes of parametric and nonparametric models. Parametric models include autoregressive moving average (ARMA) models and their modifications (Stedinger and Taylor, 1982; Papalexiou, 2018) and fractional Gaussian noise models (Mandelbrot, 1965, 1971; Mejia et al., 1972; Hosking, 1984). Nonparametric models include different bootstrap approaches (Salas and Lee, 2010; Herman et al., 2016; Srinivas and Srinivasan, 2006; Srivastav and Simonovic, 2014) and kernel density estimation (Lall and Sharma, 1996; Sharma et al., 1997). Other 395 simulation approaches for extreme events include the conditional exceedance model by Heffernan and Tawn (2004) (Keef et al., 2013; Diederen et al., 2019; Neal et al., 2013), max-stable models (Segers, 2012; Ribatet, 2013; Oesting and Stein, 2018), or copula models (Gräler, 2014; Brunner et al., 2019b). In addition to these time-domain models, there exist frequency-domain models that simulate surrogate data with the same Fourier spectra as the raw data (Theiler et al., 1992; Prichard and Theiler, 1994; Schreiber and Schmitz, 2000). Such methods are based on the randomization of the phases of the Fourier transform and 400 are known as the amplitude-adjusted Fourier transform (AAFT) (Lancaster et al., 2018; Radziejewski et al., 2000; Serinaldi and Kilsby, 2017; Brunner et al., 2019a). They have been successfully applied to simulate spatially consistent streamflow time series in multiple catchments (Brunner and Gilleland, 2020).

In addition to these statistical approaches, streamflow can be simulated using physically-based approaches. These approaches rely on a hydrological model which is driven with large ensembles of stochastically or physically generated climate input data. 405 Examples of physically-based large climate ensembles include single-model initial-condition large ensembles (SMILES; Deser et al., 2012, 2020) and reforecast simulations, i.e., forecasts generated for past periods (Hamill et al., 2006). Climate SMILEs and reforecast simulations have been used in combination with hydrological models to generate large ensembles of streamflow time series (van der Wiel et al., 2019; Willkofer et al., 2020; Brunner et al., 2021c; Brunner and Slater, 2022).

# 4 Challenges and future directions

Quantifying the frequency and magnitude of multivariate extremes is challenging for multiple reasons. Here, I discuss some of these challenges and how they could be addressed in future research.

1. **Multivariate extremes are scarce in observational records.** Therefore, frequency analyses are often associated with large uncertainties and it is challenging to study the processes governing such extreme events. To overcome the problems related to a limited sample size, simulation approaches can be used (see Section 3.4). However, these simulations need to represent different types of data features including distribution, temporal, spatial, and variable dependencies. Representing all these features simultaneously is challenging. Novel simulation approaches are needed that capture a range of different types of dependencies.

2. **Multivariate frequency analysis requires dependence modeling.** Modelling such dependence is feasible in smaller dimensions (e.g. in the bivariate setting) but becomes more complex and more computationally demanding in larger dimensions. Identifying suitable dependence structures in high-dimensions is not always straightforward and further flexible dependence structures are needed to represent temporal, spatial, and variable dependencies at the same time.

3. **Multivariate extremes are subject to change.** Extreme events are affected by various factors including land-use changes, climate, and water management (e.g. Slater et al., 2021; Blum et al., 2020; Brunner, 2021). The effects of these changes on hydrological extremes is not limited to their univariate characteristics, but extends to their dependence structure (Brunner et al., 2019c). Such non-stationarities in variable dependence, need to be accounted for in global change impact assessments.

4. **Variable dependencies need to be transferred to ungauged catchments.** Predicting the frequency and magnitude of extreme events in ungauged basins is challenging. Different methods, (i.e. regionalization approaches) are available to predict hydrological extremes or model parameters in ungauged catchments using information from gauged catchments including similarity metrics or linear and non-linear regression models. While such techniques are established in the univariate case, regionalizing multivariate extremes is more challenging because variable dependence needs to be maintained. For example, regionalizing flood peaks and flood volumes individually, may destroy the dependence between the two variables (Brunner et al., 2018; Kiran and Srinivas, 2022). Novel regionalization approaches are needed that respect such variable dependencies.

5. **Variable dependence needs to be represented in statistical and process-based models.** The representation of variable dependencies in statistical and hydrological modeling is non-trivial. For example, hydrological model simulations do neither necessarily well represent the dependence between flood peaks and flood volume (Brunner and Sikorska, 2018) nor spatial flood coherence (Brunner et al., 2021a). The representation of such dependencies in hydrological models needs to be improved by developing suitable model calibration approaches that take into account variable dependencies in addition to individual variables.

## 5 Conclusions

Multivariate hydrological extreme events can jointly affect multiple regions, occur in close temporal succession, be characterized by multiple characteristics or represent transitions from one type of extreme to another one. These different types of extreme events have in common that they involve multiple inter-related variables, whose dependence needs to be accounted for in frequency analysis and risk estimation. However, studying extreme events in a multivariate framework is challenging because of the scarceness of multivariate extreme events in observational records and the need to model variable interdependencies. Assessments of the probability and magnitude of multivariate hydrological extremes may profit from advances in the following areas: (1) the development of (stochastic) simulation approaches that represent different types of variable dependencies and allow generating large datasets; (2) the development of flexible dependence structures that represent dependencies of different strength and form; (3) and the development of hydrological model calibration procedures that enable calibrating models with respect to temporal, spatial, and variable dependencies. These method developments will facilitate change assessments for different types of multivariate hydrological extremes such as large-scale floods, successive droughts, or rapid drought-to-flood transitions. Such assessments are strongly needed in order to adapt water management strategies to future changes in impactful multivariate drought and flood events.

*Author contributions.* MIB has developed the structure and contents of this review and written, revised, and edited the manuscript.

*Competing interests.* The author is a member of the editorial board of HESS.

*Acknowledgements.* I thank the German Research Foundation (grant 2100371301) and the Swiss National Science Foundation (grant PZ00P2-201818) for supporting this writing effort.

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
