# Peer review of "Floods and droughts: a multivariate perspective on hazard estimation"

_Hydrology and Earth System Sciences, 2023_

## Author Response (AR1)

*Dear editor,*

*Thank you very much for your and the reviewers' assessment of our manuscript. I revised the manuscript according to the reviewers' comments with a particular focus on improving the discussion of different descriptors of multivariate extremes and on expanding the description of modeling approaches for multivariate extremes by including descriptions of (1) suitable univariate metrics for multivariate extremes, (2) bivariate distributions and return periods, (3) multivariate distributions, and (4) multivariate simulation approaches. I hope that you find the revised version of this manuscript suitable for publication in HESS. Thank you very much for your re-assessment.*

*Best regards,*

*Manuela Brunner*

**Reviewer 1**

The manuscript is presented as a review paper on multivariate extremes, specifically flood, and drought. The multivariate aspect of such extremes is intended in space, in time, and in their characteristics. The topic is relevant for preparedness and risk management in the current and future climates. However, the manuscript in its current form presents some limitations.

**Reply:** *Thank you very much for acknowledging the relevance of the review topic and for taking the time to provide this constructive feedback, which I address point by point below.*

The introduction on the drawbacks of the univariate approach seems in contrast with the types of multivariate extremes identified. The regional and temporal extremes fall back on a univariate approach. Indeed, they are defined based on whether, e.g., flood magnitude is above a given threshold or with a given return period at one single location. When does an extreme in one location become a multivariate extreme? How many locations should be flooded? Is the regional extent of the univariate floods an indicator of whether an extreme is multivariate or not? How so? These kinds of questions are difficult to answer from the definitions of multivariate extremes provided and it makes questioning whether it is necessary to move away from the univariate approach.

**Reply:** *Thank you for stressing the need to clarify the link between studying multivariate extremes and univariate frequency analyses. I agree that one good strategy of studying multivariate extremes is by defining univariate metrics that describe them, e.g. spatial flood extent for spatially compounding flood events. The point I would like to make here is that analyses of hydrological extreme events should go beyond focusing on one variable only and consider extreme events from a multivariate perspective. I rewrote the introduction by removing the part about univariate frequency analysis which gave the wrong impression that this tool is inappropriate to study multivariate extremes. Instead, the new introduction stresses that multivariate extremes consider more than one variable compared to univariate extremes focusing on one variable only:*

*'In July 2021, a severe and widespread flood event affected Western Germany and parts of Belgium and the Netherlands where it led to numerous fatalities and considerable damage to infrastructure (Ibebuchi et al. 2022). After such exceptional flood events, we ask: 'how frequently do such events occur?' To answer this question, one can rely on frequency*

*analyses which establish a link between the magnitude and frequency of events. Such analyses are often performed by focusing on one variable only, i.e. by taking a univariate perspective. In the case of the Germany flood, this would e.g. be flood peaks in one individual catchment. While such a focus on one variable enables the development of suitable preparedness and adaptation measures by providing magnitude and frequency estimates of extreme events, they have a major drawback: they neglect that extremes are often not univariate but multivariate phenomena, i.e. affect more than one variable.*

*To illustrate the multivariate nature of hydrologic extremes, let's again look at the 2021 flood. This flood event was not just extreme in terms of peak discharge at one location, it was also extreme in terms of the flood volume generated. Furthermore, it affected not just one catchment but multiple catchments in Germany, Belgium, and the Netherlands.*

*This example highlights that the multivariate nature of hydrological extremes can take multiple forms. In the case of peak discharge and volume, we are looking at an extreme event characterized by multiple variables and in the case of multiple affected locations at a regional extreme event. These different types of multivariate extremes have in common that they involve multiple interdependent variables, which requires a multivariate perspective. In this review, I first provide an overview of different types of multivariate hydrological extremes including regional extremes, consecutive extremes, extremes with multiple characteristics, and extremes transitions. In addition, I review tools, measures, and descriptors available to describe these different types of extremes. Second, I present modeling approaches available to model extremes in a multivariate framework, such as copula models and multivariate simulation approaches. Last, I discuss challenges related to multivariate hydrological extremes, including the regionalization of multivariate extremes to ungauged basins and the assessment of future changes in multivariate extreme events.'*

**Modification: p.1, l.14-33**

In my opinion, more emphasis should be given to the descriptors of multivariate extremes, as defined by the Author, their differences, and the implication of using one descriptor rather than another. As a matter of fact, the definition of an extreme cannot be decoupled from the descriptor used. In the manuscript, they are simply listed in tables without further implications on their use.

**Reply:** *Thank you for highlighting the need to emphasize the descriptors of multivariate extremes. I substantially expanded the description of the different descriptors and provide an overview on what types of analyses the different descriptors can be used for:*

[revised manuscript text omitted]

Section 3 on modeling multivariate extremes is about models for assessing the frequency and magnitude of multivariate hydrologic extreme events (as summarized by the Author in lines 241-243). In this section, bivariate copula models are described way more extensively compared to other methods. However, it is unclear why such a detailed description and how copula models differ from the descriptors of hydrological extremes with multiple characteristics. As a matter of fact, copulas model the dependence between two variables, where the dependence between the variables is measured by the correlation between two variables (descriptors in Table 3). It would be useful to discuss whether bivariate copulas can be applied also to regional and temporal multivariate extremes and how. Moreover, limiting the description of multivariate models to bivariate statistical methods in a review paper on multivariate extremes is not enough. I encourage the Authors to add studies and methods for higher dimensions.
**Reply:** *Thank you for stressing the need to expand the discussion of multivariate models and distributions beyond the bivariate case. I introduce bivariate copula models in detail because they are a useful tool to describe return periods in a bivariate setting, which is often used because return periods are difficult to generalize to higher than two-dimensional data. However, I fully agree that it is important to also introduce multivariate distributions and models going beyond 2 dimensions because some of the extremes discussed in this review (e.g. the spatial extremes) are higher dimensional phenomena. Therefore, I substantially expanded section 3 (Modeling multivariate extremes) by including descriptions of (1) suitable univariate metrics for multivariate extremes, (2) bivariate distributions and return periods, (3) multivariate distributions, and (4) multivariate simulation approaches.*

*'**Univariate metrics for multivariate extremes:** Different approaches have been developed to quantify the frequency of multivariate extremes. The easiest work around for dealing with multivariate extremes is to describe the complex phenomena with a suitable univariate descriptor, such as describing regional floods by flood extent. Such univariate descriptors can be used in a univariate frequency analysis to determine the frequency and magnitude of events. Such a univariate frequency analysis first defines a sample of extreme events using either a block maxima/minima or a peak-over-threshold/threshold-level approach (Meylan et al. 2012). Second, it fits a suitable theoretical distribution to the sample of extreme events. In the case of block maxima, one usually works with a Generalized Extreme Value (GEV) distribution and in the case of threshold exceedances with a Generalized Pareto distribution (GPD) (Coles 2001). The goodness-of-fit of the distribution chosen is assessed using a test for*

*extreme values such as the Anderson--Darling or Cramér-von-Mises test (Laio et al. 2004). Once a suitable distribution has been identified, one can use the probability distribution function to determine the probability of occurrence of a certain event or the quantile function to determine the magnitude of an event with a certain non-exceedance probability or return period (Figure 6). The relationship between the non-exceedance probability p and the corresponding return period T is expressed as follows:*

$$T = mu/(1-p),$$

[revised manuscript text omitted]

**Modification: p.15, l.350-377**

Point-by-point comments:

Line 28 and Line 280: my suggestion is to cite textbooks or the original journal papers where these concepts are first defined. For example, G. Salvadori and C. De Michele earlier works.

**Reply:** *Thank you for this suggestion. I included a few of the original journal papers, where the return period concept was introduced, and the paper by Salvadori and De Michele.*

**Modification: p.13, l.293**

Line 88: "precipitation dependence" dependence to what?

**Reply:** *I specified that I was referring to 'precipitation spatial dependence'.*

**Modification: p.4, l.75**

Figure 4: it would help to have more information on how drought is defined

**Reply:** *I specified that 'droughts were here defined using a variable threshold at the 15th flow percentile.'*

**Modification: p.6, caption Figure 3**

Lines 182 – 200: discussion about variables dependence is a bit vague. Which variables? Is it a bi-variate dependence? The example of dependence between peak and volume for hydraulic design should be elaborated further.

**Reply:** *Thank you very much for pointing out the need for clarification. I provide a few examples of variable pairs of interest, specify that we are talking about bivariate dependence, and use the example of peak-volume to illustrate the importance of considering bivariate/multivariate relationships in hydraulic design by adding the following sentences: 'For example, flood duration and volume or flood volume and flood peak show strong correlations (Figure 5), i.e. they show bivariate dependence. [...] Such bivariate interdependence is e.g. found for drought deficit and duration or drought deficit and intensity (Figure 5a,b).[...] Such dependence, e.g. between peak discharge and flood volume, is*

*important for hydraulic design because dam failure depends not only on flood peak but also volume (De Michele et al. 2005).*
**Modification: p.9, l.210-211**

Line 321: studies in higher dimensions should be added to the manuscript
**Reply:** *I rewrote Section 3 (Modeling multivariate extremes) and added a subsection called 'multivariate distributions', which discusses different models for multivariate extremes including conditional exceedance models, max-stable models, the multivariate generalized Pareto distribution, and high-dimensional copula models such as vine copulas.*
**Modification: p.12-16, Sections 3.1-3.3**

**Reviewer 2**

This manuscript presents a review of some hydrological problems that can be characterized in terms of a multivariate extreme value distribution. The identified hydrological conditions that require a probabilistic estimation in terms of event magnitude and occurrence are listed and briefly discussed along with the metrics that can be generally used to determine the dependence among the variables. Further, copula for multivariate frequency analysis and continuous time serie simulation are introduced as strategies for modeling those phenomena. Due to the variety and complexity of the problems mentioned in the review paper, each of them is only hinted at, missing an in-depth discussion about several important issues. Further, many interesting works about multivariate statistical modeling are not mentioined at all; indeed, also the most recent literature on the topic is very rich. Based on these consideration, I suggest the Author to revise her work trying to improve the description of the phenomena, especially those problems that are still unsolved, and enlarge the state of the art description referring the interested readers to the most recent papers (and books) that provide well established and innovative solutions with a deeper insight into the mentioned problems.

**Reply:** *Thank you very much for your assessment and for highlighting that the discussion of multivariate statistical models was too superficial in the first version of the manuscript. I rewrote Section 3 (Modeling multivariate extremes) by including multivariate statistical models going beyond copula approaches, which are applicable to higher dimensional problems such as spatial extremes. Furthermore, I substantially expanded the description of metrics used to describe the four types of multivariate hydrologic extremes I focus on in this review, i.e. regional extremes, consecutive extremes, extremes with multiple characteristics, and extremes transitions.*

**Modification: p.12-16, Sections 3.1-3.3**

---

## Author Response (AR2)

*Dear Dr. Hrachowitz,*

*Thank you very much for your re-assessment. I appreciate the further comments by Reviewer #1 and address them in the revised version of the manuscript.*

*Best regards,*

*Manuela Brunner*

**Reviewer 1**

I would like to thank the Author for addressing the comments provided very carefully. The manuscript has improved substantially.

**Reply:** *I would like to thank the reviewer for their careful reassessment.*

At the same time, the issue of identifying multivariate extremes is still highly linked to identifying univariate extremes and then looking at them together. Questions such as When does an extreme in one location become a multivariate extreme? How many locations should be flooded? Is the regional extent of the univariate flood an indicator of whether an extreme is multivariate or not? are missing. This is the trickiest aspect of multivariate extremes and should be addressed. Moreover, starting from the premise that environmental variables are connected because the system is connected, when is a "simpler" univariate approach not anymore sufficient?

**Reply:** *Thank you very much for pointing out the need to further elaborate on the definition of regional extremes and their relationship to spatial extent. I clarified that 'Here, we talk about regional extremes as soon as a local perspective is no longer sufficient, i.e. when floods have a larger spatial extent and more than one catchment is affected, which requires a multivariate perspective.'*

Finally, another point missing and worth addressing is whether the use of a descriptor already implies the choice of a modeling approach. For example, using dependence metrics, e.g., Kendall's tau, already implies the use of models based on bivariate copulas. In other words, it would be useful to explicitly address the issue of choosing a descriptor and a model and whether the choice of one already implies the choice of the other.

**Reply:** *Thank you for raising this point. No, the choice of descriptor does not necessarily imply the choice of a modeling approach. One could use Kendall's tau do describe pairwise dependencies and then use a stochastic model to simulate streamflow/extremes at a pair of stations.*

Lines 124-125: It seems something is missing in this sentence.

**Reply:** *Thank you for highlighting the need for rephrasing. The revised sentence reads: 'For example, the synchrony scale measures over which distance around a catchment, multiple rivers experience flooding at the same time.'*